# Anti-obesity compounds, Semaglutide and LiPR, and PrRP do not change the proportion of human and mouse POMC+ neurons

Sara K. M. Jörgensen[1], May Surridge-Smith[1¤], Kimberley Jones[1], Lenka Maletínská[2], Nicholas D. Allen[1], David Petrik [1]*

1 School of Biosciences, Cardiff University, Cardiff, Wales, United Kingdom, 2 Institute of Organic Chemistry and Biochemistry of the Czech Academy of Sciences, Prague, Czech Republic

¤ Current address: School of Medicine, Dentistry and Biomedical Sciences, The Queen's University of Belfast, Belfast, Northern Ireland, United Kingdom
* petrikd@cardiff.ac.uk

## Abstract

Anti-obesity medications (AOMs) have become one of the most prescribed drugs in human medicine. While AOMs are known to impact adult neurogenesis in the hypothalamus, their effects on the functional maturation of hypothalamic neurons remain unexplored. Given that AOMs target neurons in the Medial Basal Hypothalamus (MBH), which play a crucial role in regulating energy homeostasis, we hypothesized that AOMs might influence the functional maturation of these neurons, potentially rewiring the MBH. To investigate this, we exposed hypothalamic neurons derived from human induced pluripotent stem cells (hiPSCs) to Semaglutide and lipidized prolactin-releasing peptide (LiPR), two anti-obesity compounds. Contrary to our expectations, treatment with Semaglutide or LiPR during neuronal maturation did not affect the proportion of anorexigenic, Pro-opiomelanocortin-expressing (POMC+) neurons. Additionally, LiPR did not alter the morphology of POMC+ neurons or the expression of selected genes critical for the metabolism or development of anorexigenic neurons. Furthermore, LiPR did not impact the proportion of adult-generated POMC+ neurons in the mouse MBH. Taken together, these results suggest that AOMs do not influence the functional maturation of anorexigenic hypothalamic neurons.

## Introduction

Obesity remains one of the biggest medical and socioeconomic challenges. It increases the risk of Type 2 diabetes [1], cardiovascular diseases [2], or cancer [3,4] and it is the second leading cause of disability [5], resulting in a massive economic impact of pandemic proportions [6,7]. The primary cause of obesity is impaired energy homeostasis caused by reduced energy expenditure and excessive food

**Data availability statement:** All relevant data for this study are publicly available from the OSF repository (https://doi.org/10.17605/OSF.IO/84ZXK).

**Funding:** This work was funded by UK Academy for Medical Sciences (SBF007\100124 to D.P.) and Wales Gene Park (HCRW RDI2020 to N.D.A.).

**Competing interests:** The authors have declared that no competing interests exist.

intake [8]. To curb the development and maintenance of obesity, pharmacological regulation of appetite and feeding behaviour has emerged as one of the most used therapeutic approaches [9]. The most effective AOMs that reduce body weight by decreasing food intake are chemically modified analogues of anorexigenic neuropeptides. In 2023, Semaglutide (Ozempic/Wegovy), a palmitoylated Glucagon-like Peptide-1 (GLP-1) analogue approved as an AOM [10], ranked as the most prescribed new drug in the world [11]. In parallel to existing medications, novel compounds have been investigated as potential AOMs. One of these compounds, LiPR, a lipidized analogue of the Prolactin Releasing Peptide (PrRP), produces a fast and robust reduction of body weight in mice [12–14].

Hypothalamus, a ventral part of diencephalon responsible for regulating basic physiological functions, is also critical for controlling energy homeostasis [15,16]. In the MBH, anorexigenic (i.e., food-intake suppressing) and orexigenic (i.e., food-intake promoting) neurons reside in primary nutrient and hormone sensing Arcuate (Arc) and secondary Dorso-Medial (DMN) and Para-Ventricular (PVN) nuclei [17]. These neurons sense metabolites and satiety signals from the periphery and regulate eating behavior and appetite [18–21]. In Arc, feeding is suppressed by POMC+ neurons [18] and stimulated by Neuropeptide Y (NPY+)/Agouti-related protein (AgRP+) neurons [19,20], which are inhibited by DMN neurons [22]. Most of these neurons originate during embryogenesis. However, some MBH neurons are newly generated during adulthood from resident adult neural stem cells in the process of adult neurogenesis. These new MBH neurons are critical for energy homeostasis [23] as evidenced by their ablation, which exacerbates obesity [24,25].

We recently showed that LiPR and Liraglutide, another analogue of GLP-1, promote hypothalamic adult neurogenesis in mice and increase the number of new, adult-generated hypothalamic neurons in the context of the diet-induced obesity [26]. However, we did not determine the phenotype of these new hypothalamic neurons. Our results also showed for the first time that PrRP activates hypothalamic peptidergic neurons derived from hiPSCs [26]. These findings suggest that AOMs increase survival of new hypothalamic neurons by cell-intrinsic actions, however, they did not determine AOM mechanism of action, with one possibility being that they act during differentiation and change neuronal phenotypic maturation. To address this, we exposed hiPSC-derived neuronal progenitors during their maturation to PrRP or Semaglutide and determined whether this treatment would change the proportion of POMC+ neurons. We hypothesized that AOMs would increase the proportion of POMC+ neurons as a part of their long-term anorexigenic action to rewire the hypothalamus. Contrary to this expectation, however, we report here that neither PrRP nor Semaglutide changed the proportion of POMC+ hiPSC-derived neurons. Moreover, we analysed differentiation *in vivo* and did not observe any differences in the proportion of POMC+ adult-generated neurons in the MBH of mice treated with LiPR. Taken together these findings suggest that PrRP/LiPR or Semaglutide do not alter the proportion of anorexigenic hypothalamic neurons.

## Methods and materials

### Anti-obesity compounds

The following three anti-obesity compounds were used. The full-length, 31 amino acid human Prolactin Releasing Peptide (hPrRP31): SRTHRHSMEIRTPDINPAWYASRGIRPVGRF-NH2 (Molecular Weight, M.W., 3661.9). A palmitoylated analogue of hPrRP31, LiPR: SRTHRHSMEIK (N-γ-E(N-palmitoyl)) TPDINPAWYASRGIRPVGRF-NH$_2$ (M.W. = 4004). LiPR was synthesized and purified at the Institute of Organic Chemistry and Biochemistry, Czech Academy of Sciences, Prague (CAS), Czech Republic, as previously described [27]. Semaglutide: H-Aib-EGTFTSDVSSYLEGQAAK (PEG2-PEG2-γ-Glu-17-carboxyheptadecanoyl) EFIAWLVRGRG (M.W. = 4113.6). Semaglutide was synthesized and purified by BOC Sciences, Inc., New York, USA. All compounds were kept at −20 °C until reconstituted. Stock solution of 1 mM hPrRP31 or 0.25 mM Semaglutide in sterile cell-grade water were kept at 4 °C and added to cell media to the final concentration of 1 µM. hPrRP31 or Semaglutide were added with every media change during the maturation period. LiPR was administered to mice – more details below.

### hiPSC lines

Three hiPSC lines were used. Two are widely available from cell repositories: HIPSC-Kolf (Kolf-C1), https://www.ebi.ac.uk/biosamples/samples/SAMEA2398402;, SAMN08388588: CS25iCTR18n (18n6), https://www.ebi.ac.uk/biosamples/samples/SAMN08388588. One line, i900, was generated in the laboratory of Meng Li, Cardiff University, United Kingdom. Kolf-C1 and 18n6 lines were derived from skin fibroblasts of 46–75 years old males. i900 was derived from skin fibroblasts of 25–30 years old female. Before differentiation in our experiments, the hiPSC lines had undergone the following number of passages: 28 for Kolf-cl: 26 for 18n6, 31 for i900.

### hiPSC differentiation

hiPSC lines were cultured until ~80% confluency and then replated for neuronal differentiation into peptidergic hypothalamic neurons. The differentiation lasted from day 0 (the beginning of differentiation) until day 14 when a proportion of cells were harvested for quality control and harvesting of mRNA. After the differentiation, cells were matured with BDNF from day 14 to day 30. During this maturation period, cells were exposed to human full-length (31 amino acid long) PrRP (hPrRP31) or Semaglutide diluted in sterile cell culture grade water. As a control, cells were exposed to water. After maturation at day 30, cells were processed for immunocytochemistry or harvested for mRNA.

### hiPSC culture

hiPSCs were kept in mTesR medium (Stem Cell Technologies (SCT) 85850) and incubated at 37 °C, 20% O$_2$, 5% CO$_2$. Cells were grown on 6-well plates coated with vitronectin (Thermo Fisher Scientific (Thermo), A14700), diluted 1:1000 in DPBS without Calcium and Magnesium (DPBS(-/-)) (Thermo 14190144). When cultures approached ~80% confluency, they were passaged. mTesR was replaced by ReLeSR (SCT 100-0484) and incubated for 2 minutes at 37 °C. Then, cells were gently washed off the plates with mTesR, centrifuged (3 minutes, 1000 rpm) and plated with fresh mTesR.

### Hypothalamic neural progenitor differentiation

The differentiation of hiPSC into peptidergic hypothalamic neurons was performed as described previously [28,29] with the only modification: XAV939 was replaced by IWR1 (both being Tankyrase (Wnt pathway) inhibitors). hiPSCs (~ 80% confluent) were replated onto plates coated with Geltrex (Thermo A1413201) and fed neuronal differentiation medium based on N2B27 and small molecules. The N2B27 medium consisted of the following: Neurobasal™-A Medium (500 ml, SCT 10888022), DMEM/F-12, GlutaMAX™ Supplement (500 ml, SCT 31331093), GlutaMAX™ Supplement (10 ml, SCT 35050038), B-27™ Supplement (20 ml, SCT 17504044), N-2 Supplement (5 ml, SCT 17502048), L-Ascorbic acid (100

mg, Thermo A4403), Sodium Bicarbonate (10 ml, 7.5% solution, SCT 25080094), Penicillin-Streptomycin (10 ml, 10,000 U/ml, SCT 15140122), MEM Non-Essential Amino Acids Solution (5 ml, SCT 11140035). The N2B27 medium with small molecules was fed to cells in the following timeline. Day 0: N2B27 + 1.5µM IWR1 (SCT 72564), 100 nM LDN-193189 (SCT 72147), 10 µM SB431542 (SCT 72232); Day 2: Day 0 medium + 1 µM SAG (SCT 73414), 1 µM Purmorphamine (SCT 100-1049); Day 4: N2B27 + 1µM IWR1, 75 nM LDN-193189, 7.5 µM SB431542, 1 µM SAG, 1 µM Purmorphamine; Day 6: N2B27 + 0.5 µM IWR1, 50 nM LDN-193189, 5 µM SB431542, 1 µM SAG, 1 µM Purmorphamine; Day 8: N2B27 + 0.25 µM IWR1, 25 nM LDN-193189, 2.5 µM SB431542, 5 µM DAPT (SCT 72082); Day 10 and 12: N2B27 + 5 µM DAPT; Day 12: N2B27 + 5 µM DAPT. Full medium change were performed daily At Day 14, following differentiation, cultures were incubated in Accutase cell detachment medium (Stem Cell Technologies, 07922, 07920) at 37 °C for 10 minutes. Dissociation was terminated by dilution with N2B27 medium (10µM Y-27632), and cells gently triturated before harvesting by centrifugation for 3 minutes at 200g. For cryopreservation cells were resuspended in CryoStor ® (Stem Cell Technologies, 100-1061) and frozen at −80 °C.

## Neuronal maturation

Neuronal maturation followed an established protocol [28,29]. Cells were resuspended in N2B27 (with 10 µM Y-27632, 5µl DAPT) in T25 flasks coated with Geltrex and fed with N2B27 (with 5µl DAPT) for three days before harvesting as above and being replated for maturation. For ICC, $10^5$ cells were plated per cm² on glass coverslips coated with Poly-D-Lysine (PDL, Thermo A3890401) followed by Geltrex. For qPCR experiments, cells ($10^5$ per cm²) were plated in 12-well plates coated with Geltrex. Cells were matured in N2B27 media with BDNF (SCT 78005), which was added at each feeding to a final concentration of 10 ng/ml. Cells were fed every 2 days with 75% of the media volume changed at each feed.

## RT-qPCR

RNA extraction from thawed cells was performed with the RNeasy Mini plus kit (Qiagen, UK) according to the manufacturer's instructions. RNA was re-transcribed into cDNA using Superscript™ III polymerase (Thermo 18080093) with random primers (Thermo 48190011) and RNase inhibitor (Thermo 10777019). qPCR reaction was performed using Fast SYBR Green dye (Thermo) in technical duplicates for each sample on a Step One Plus Real time PCR system (Life Technologies) to determine relative expression of selected mRNA transcripts (denaturation for 5 min at 95 °C, followed by 45 cycles of 95 °C denaturation and 60 °C annealing and elongation). The results were analysed using the AccuSEQ software (Life Technologies) with the amplification cycle (Ct) values determined by maximum 2nd derivation method and following the $2^{-\Delta Ct}$ method [30] with normalization to the expression levels of GAPDH. The sequences of used primers (250 nM) were designed by using the Primer–BLAST online tool (NCBI-NIH): *GAPDH* forward = GCACCGTCAAGGCTGAGAAC, *GAPDH* reverse = TGGTGAAGACGCCAGTGGA; *PRLHR* forward = TTTGGATCCGTTCAGCTCCC, *PRLHR* reverse = GGAAGTTCGTCACGTTGTGC; *NPFFR2* forward = TCCTCAGTTGCGAAATTAGGATGT, *NPFFR2* reverse = CCACTTGAGGCTGGTGAAGA; *PRLH* forward = GCACCCCTGACATCAATCCT, *PRLH* reverse = AGCCATCCTGGGACGACATA; *GLP1* forward = GAGGAAGGCGAGATTTCCCA, *GLP1* reverse = CCCTGGCGGCAAGATTATCA; *NPY* forward = GCTGCGACACTACATCAACCTC, *NPY* reverse = CTGTGCTTTCTCTCATCAAGAGG; *POMC* forward = CTCACCACGGAAAGCAACC, *POMC* reverse = CTGCTCGTCGCCATTTCC; *TBX3* forward = GACACTGGAAATGGCCGAAG, *TBX3* reverse = CTGCTTGTTCACTGGAGGAC; *PRDM12* forward = GCACGTAACGAACAGGAGCA, *PRDM12* reverse = GTGAGTTTCCGTACCACACCA; *AGRP* forward = TGCAGAACAGGCAGAAGAGG; *AGRP* reverse = GCAGGACTCATGCAGCCTTA; *CARTPT* forward = ATGATGCCTCCCACGAGAAG, *CARTPT* reverse = CCTTTCCTCACTGCACACT; *MC3R* forward = AACACTGCCTAATGGCTCGG, *MC3R* reverse = GTTTTCCAGCAGACTGACGATG; *INSR* forward = GCAACATCACCCACTACCTGGT, *INSR* reverse = GAATGGTGGAGACCAGGTCCTC; *LEPR* forward = GCAGTCTATGCTGTTCAGGTG, *LEPR* reverse = CCAAAATTCAGGTCCTCTCATAGG; *CCKAR* forward = CGCTTTTCTGCTTGGATCAG, *CCKAR* reverse = CTTGTTCCGAATCAGCACG; *TTR*

forward = CGTGCATGTGTTCAGAAAGGC, *TTR* reverse = CTCCTCAGTTGTGAGCCCATG; *NKX2-1* forward = ACTC-GCTCGCTCATTTGTTG, *NKX2-1* reverse = GGAGTCGTGTGCTTTGGACT; *FOXJ1* forward = GACCTACTCCCTCAAC-CCCT, *FOXJ1* reverse = GCTGCTCTGCGAAGTCATTG.

## Immunocytochemistry and confocal imaging

Cells were fixed with 4% paraformaldehyde (PFA) in DPBS(-/-). Cells were stained in 1X PBS with 10% donkey serum (Sigma-Aldrich D963) and 0.1% Triton X-100 (Sigma-Aldrich, T8787). After blocking, cells were incubated in the carrier with primary antibodies overnight at 4 °C. After washout, the cells were stained with secondary antibodies (1:300) for 2 h at room temperature and nuclei stained with DAPI (1:1000, Roche, Munich, Germany). Cells were mounted with ProLong Diamond Antifade Mountant (Thermo P36970) onto glass slides. Following primary antibodies were used: monoclonal antibody to ACTH: A1H5 to stain for POMC (1:5000, developed and kindly shared by Prof. Anne White, University of Manchester, UK), chicken anti-MAP2 monoclonal antibody (1:500, Abcam ab92434), DAPI ready-made (Sigma-Aldrich MBD0015).

## Cell quantification

Cells were imaged on a laser scanning confocal microscope Zeiss LSM 780 at 20X magnification for cell quantification and 63X magnification for morphological analysis and selected representative images. For each hiPSC line, cells were grown in technical triplicates for both control and treatment groups and each technical replicate was imaged for a minimum of 2 confocal Regions of Interest (ROI) scans used for the analysis. The proportions of POMC+ cells from each technical replicate were averaged and expressed per individual hiPSC line. On average, around 200 MAP2-positive (MAP2+) cells were analysed per confocal scan by a researcher blind to the experimental groups. Manual quantification was done either in Fiji ImageJ software [31] or in Zeiss ZEN Blue software. Automated quantification of DAPI+ nuclei was conducted using an automated particle analysis macro pipeline in Fiji ImageJ. On average, 1500 DAPI+ nuclei were identified per confocal scan. Morphological analysis of MAP2 + POMC+ cells was performed on Fiji ImageJ using the "simple neurite tracer" plugin [32]. Using the "measure" function in ImageJ, the mean pixel density of POMC, MAP2 and DAPI signal was determined from the same confocal Z stack composites used for the cell quantifications. The primary neurites extending from the soma were traced until branches bifurcated or were not traceable due to crossing with other cell processes. The number of branches were manually quantified. Representative images were generated by compressing the confocal Z stack into a single plain in the Zeiss ZEN Blue software.

## Animal treatment and immunohistochemistry

All animal experiments followed the ethical guidelines for animal experiments and the Act of the Czech Republic Nr. 246/1992 and were approved by the Committee for Experiments with Laboratory Animals of the Academy of Sciences of the Czech Republic. C57BL/6 male mice that were 4 weeks old were obtained from Charles River Laboratories (Sulzfeld, Germany). The mice were housed under controlled conditions at a constant temperature of $22 \pm 2°C$, a relative humidity of 45–65% and a fixed daylight cycle (6 am–6 pm), with 5 mice per cage. The animals were provided free access to water and the standard rodent chow diet chow (Ssniff Spezialdiäten GmbH, Soest, Germany, #V1535 RM-Haltung). Mice were administered BrdU (1 mg/ml) in sterile tap water with 1% sucrose for first 5 days followed by 16 days of post-BrdU chase. Mice were administered LiPR (5 mg LiPR/kg of body weight) for 21 days by a daily single subcutaneous (s.c.) injection [26]. As a control, LiPR vehicle was injected s.c. After 21 days of treatment, animals were sacrificed by pentobarbital overdose followed by a complete exsanguination, transcardially perfused with 4% PFA and brain isolated as described before [26]. The total anaesthesia was used to nullify suffering. The brains were isolated and postfixed in 4% PFA for overnight. The PFA was replaced by 30% sucrose and brains let to sink at 4 °C. 40 µm coronal sections (spanning entire

hippocampus) were cut in serial sets of 12 for stereological evaluation. Slide-mounted immunohistochemistry (IHC) with IHC carrier (1×PBS, 0.5% Triton X-100, 5% normal donkey serum) used the HCl pre-treatment and anti-ACTH (1:500) and anti-MAP2 (1:300) antibodies described above and a rat anti-BrdU antibody (1:400; BioRad MCA6144) as described previously [26]. Brain sections were incubated in primary antibodies overnight at RT and with secondary antibodies (1:300) for 1.5 hours at RT. Sections were counterstained with DAPI (1:1000; Roche) and cover-slipped using the ProLong Diamond antifade mountant. Quantification of marker-positive cells in the MHB (bregma −1.2 to −2.3 mm) was performed stereologically in a 1/12 sampling fraction as described before [26]. Z-stack images were obtained in the Zeiss ZEN Blue software using 20X apochromatic objectives on the Observer.Z1 Zeiss LSM780 confocal microscope by an observer blind to the experimental groups. Density and absolute number of BrdU+Map2+neurons expressing POMC were quantified in the MBH (including the ME) as described previously [26]. In addition to cell quantification, pixel density of POMC signal was determined in the MBH or Arc delineated in images of confocal Z stack composites using ImageJ "measure" function.

## Statistical analysis

Numbers of biological and technical repeats are provided in the methods above. Datasets were analysed with Microsoft Excel and GraphPad Prism. The statistical analysis has adhered to the following procedural algorithm. First, statistically significant outliers were identified by the Grubb's test (ESD method, alpha=0.05) and removed from following statistical analyses. Second, the normality of data distribution was determined by D'Agostino & Pearson's or Kolmogorov–Smirnov normality tests. Data presented in this manuscript were normally distributed and analysed by parametric statistical tests. For simple comparison of two data groups, the un-paired two-tailed *T*-Test was used. For multiple factor or group comparison, One-way analysis of variance (ANOVA) was used with the Bonferroni's or Tukey's post-hoc test for the cross-comparison of individual data sets. The data were presented as mean±standard error of mean (SEM). Results were considered significant with $P<0.05$ (one asterisk). Two asterisks represent values of $P<0.01$, three asterisks represent $P<0.001$.

## Results

### Exposure to PrRP during neuronal maturation does not alter the proportion of POMC+ cells or expression of selected genes

We generated peptidergic hypothalamic hiPSC-derived neurons using previously established protocols [28,29]. After two weeks of differentiation, hiPSC-derived neurons were exposed to hPrRP31 (further referred to as PrRP) during neuronal maturation to determine its effects on phenotypic maturation and gene expression (Fig 1A). hPrRP31 was used instead of LiPR because it is not plastic adhesive. This protocol generated morphologically mature MAP2+neurons clearly expressing POMC in cell soma and processes (Fig 1B) in all 3 hiPSC lines (Fig 1C–H). Cell quantification of the proportion of POMC+MAP2+neurons revealed no statistically significant difference between Control and PrRP treated cells (Fig 1I). There was however a large variability in the proportion of POMC+MAP2+neurons between the 3 hiPSC lines, with i900 showing almost 5-fold higher proportion than Kolf-C1 or 18n6 (Fig 1J). However, despite these cell line-intrinsic differences, there were no differences in the proportions of POMC+/MAP2+neurons between Control and PrRP treated cultures of the individual lines. To further analyse effects of PrRP, we quantified the proportion of POMC+/DAPI+ cells but again found no statistically significant differences between Control and PrRP treated groups (S1 Fig). This suggests that the exposure to PrRP during neuronal maturation does not change the proportion of anorexigenic POMC+ neurons.

Next, we determined if PrRP exposure changes expression of selected genes at the end of neuronal differentiation (day 14) and at the end of maturation (day 30) of Control and PrRP-treated neurons. Using Real Time quantitative PCR (RT-qPCR), we observed that neurons express transcription factors NKX2.1 and FOXG1, which are critical for the diencephalon and forebrain development [29], respectively (S1 Fig). Consistent with successful neuronal maturation, *NKX2.1* and *FOXG1* were both significantly downregulated between day 14 to day 30, in both Control and PrRP-treated cultures. The

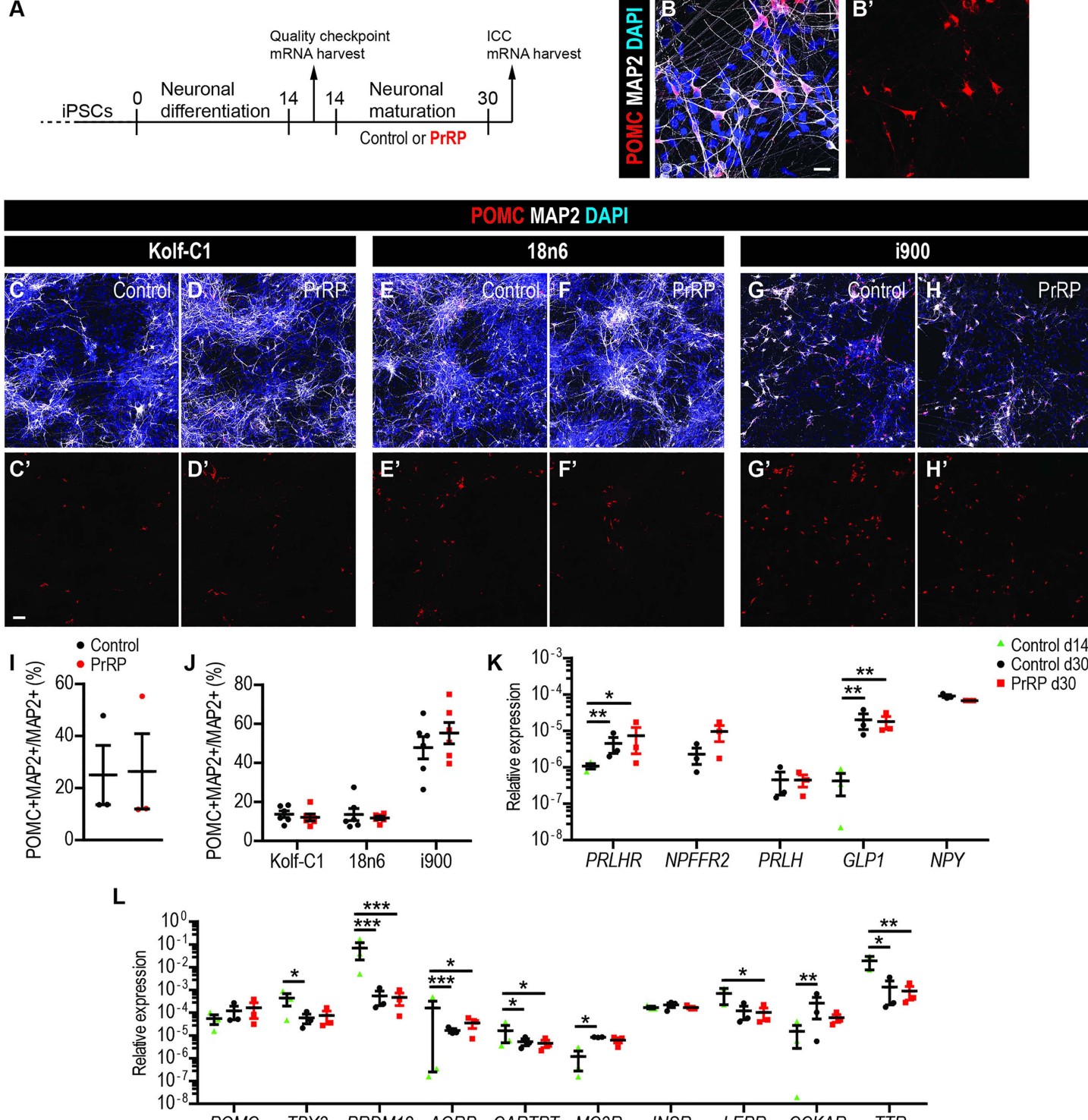

**Fig 1. PrRP does not alter the proportion of POMC+ hiPSC-derived neurons (A) A schematic of the experimental timeline.** (B) Representative confocal images of hiPSC-derived neurons stained as indicated at day 30. (C-H) Representative confocal images of neurons derived from three hiPSC lines and treated with the vehicle (Control, C, E, G) or PrRP (D, F, H) at day 30. (I) Quantification of the POMC+MAP2+/MAP2+ proportion. (J) Quantification of the POMC+MAP2+/MAP2+ proportion per hiPSC line (individual data points represent technical replicates). (K-L) RT-qPCR expression of selected genes. Scale bar (s.b.) = 20 μm (B), 50 μm (C-H). n = 3 hiPSC lines. Two-tailed T-Test: *$p < 0.05$, **$p < 0.01$, ***$p < 0.001$. Data are presented as mean ± SEM.

neurons also expressed genes for PrRP receptors, *PRLHR* (GPR10 receptor) and *NPFFR2*, and maturation increased the expression of *PRLHR* but with no effect of PrRP exposure (Fig 1K). In addition, neurons expressed other orexigenic and anorexigenic markers such as *PRLH* (encoding PrRP), *NPY* and *GLP1* (encoding the Glucagon-like Peptide 1, GLP-1), with upregulation during maturation, but with no effect of PrRP treatment (Fig 1K). To further explore the effects of maturation and PrRP on the neurons, we determined expression levels of selected genes that are important for neuronal identity and energy homeostasis (Fig 1L). The neurons expressed *POMC* and two transcription factors essential for the development of POMC+ neurons, *TBX3* and *PRDM12* [33]. Reduction of their expression from day 14–30 suggests cells underwent neuronal maturation but there was no difference between Control and PrRP-treated cells at day 30. Besides POMC, the neurons also expressed other genes of MBH neuronal identity [34] such as orexigenic *AGRP* (encoding AgRP), anorexigenic *CARTPT* (encoding Cocaine- And Amphetamine-Regulated Transcript Protein), which were reduced in expression during maturation and anorexigenic MC3R (encoding the melanocortin 3 receptor), which was increased during maturation in Control. Finally, the neurons expressed genes important for nutrient sensing and energy homeostasis signalling such as *INSR* (encoding the insulin receptor), *LEPR* (encoding the leptin receptor)[35], *CCKAR* (encoding the cholecystokinin A receptor)[36] and TTR, encoding transthyretin, a protein involved in the expression and distribution of the thyroid hormone, that had its expression upregulated in the MBH of mice treated with LiPR [26,37]. Neuronal maturation from day 14–30 reduced expression of CCKAR and increased expression of MC3R, and maturation with PrRP reduced expression of *LEPR.* However, there was no difference in expression in any of the tested genes at day 30 between Control and PrRP groups suggesting that PrRP does not alter how maturing neurons regulate key identity or energy homeostasis-related genes.

## PrRP does not change neuronal morphology or POMC expression

Next, we further explored potential effects of PrRP exposure on neuronal morphology or levels of POMC in matured neurons at day 30. Because neurons were overlapping and convoluted, we could not reliably trace the processes of individual neurons beyond the first bifurcation or perform the Scholl analysis. However, we quantified the number of primary MAP2 + neurites in POMC+ cells and found no statistically significant difference between Control and PrRP groups (Fig 2A–B). To further determine whether PrRP changed expression of POMC, we quantified the POMC pixel density in images of day 30 neurons. We did not find any difference in the ratio of POMC/DAPI pixel density or in POMC mean pixel density between Control and PrRP groups (Fig 2C–D). These results suggest that exposure to PrRP during neuronal maturation does not change cell morphology or expression of POMC.

## Exposure to Semaglutide during neuronal maturation does not alter the proportion of POMC-expressing neurons

Because PrRP failed to change the proportion of POMC neurons during neuronal maturation and because we previously observed that GLP-1 and PrRP analogues share some of their effects on adult neurogenesis [26], we exposed maturing

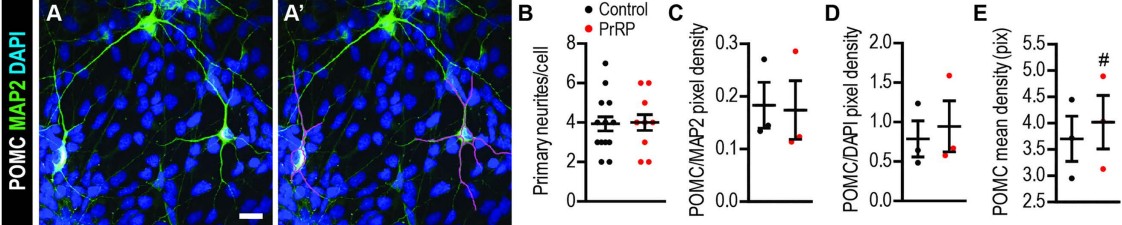

**Fig 2. PrRP does not change the number of neurites or POMC expression.** (A) A representative confocal image of POMC+ neurons traced for MAP2 + neurites. (B) Quantification of the number of primary neurites per cell. (C) The ratio of the mean pixel density for POMC and MAP2 in hiPSC-derived neurons. (D) The ratio of the mean pixel density for POMC and DAPI in hiPSC-derived neurons. (E) The mean pixel density of POMC in hiPSC-derived neurons. S.b. = 20 μm. n = 3 hiPSC lines. Two-tailed T-Test: #$p < 0.06$. Data are presented as mean ± SEM.

neurons from day 14–30 to Semaglutide (Fig 3A–B). Like our PrRP experiments, Semaglutide failed to change the proportion of POMC+ neurons (Fig 3C–D) suggesting that it does not influence the phenotypic maturation of anorexigenic hypothalamic neurons.

### LiPR does not change the proportion of new POMC+ neurons in the mouse hypothalamus

To determine whether PrRP signalling can influence the development of new hypothalamic neurons *in vivo*, we treated adult mice with vehicle (Control) or LiPR, which can pass through the Blood Brain Barrier [14], for 3 weeks. Concurrently, mice were administered BrdU, a thymidine analogue, which is incorporated into DNA of dividing cells, for 5 days in drinking water followed by 16 days without BrdU, so-called chase (Fig 4A). This strategy allowed us to label new adult-generated neurons in the MBH as we did previously [26] and stain brain sections for POMC (Fig 4B–C). Stereological analysis revealed no difference between Control and LiPR-treated mice in the absolute number (Fig 4D) or density (Fig 4E) of BrdU+ cells with or without POMC in the MBH. This suggests that LiPR does not alter the neuron subtype specification in hypothalamic adult neurogenesis. To address whether LiPR changed expression of POMC, we quantified the pixel density of POMC signal. There was a statistically significant lower POMC pixel density in both the Arc and MBH of LiPR-treated mice (Fig 4F), suggesting lower expression of POMC in both adult and embryo-generated neurons.

## Discussion

Our previously published results suggest that AOMs promote adult neurogenesis [26]. Given the widespread use of Semaglutide [11], we wanted to address whether AOMs exhibit their anorexigenic effects by increasing the proportion of new POMC+ neurons. We hypothesized that AOMs have a potential to skew the neuronal phenotypic maturation as a part of their long-term effects. To our knowledge, we were the first that attempted to determine this both in the hiPSC-derived neurons and in mice. Contrary to our hypothesis, our results jointly suggest that neither PrRP or GLP-1 analogues change the proportion of POMC+ neurons derived from hiPSCs or from adult neural stem cells in mouse MBH. The protocol used for hiPSC hypothalamic neural differentiation clearly generated POMC-expressing neurons, and their maturation altered expression of genes relevant for anorexigenic or orexigenic functions or metabolism. However, the exposure to PrRP or Semaglutide did not change their expression or neuronal morphology. Similarly,

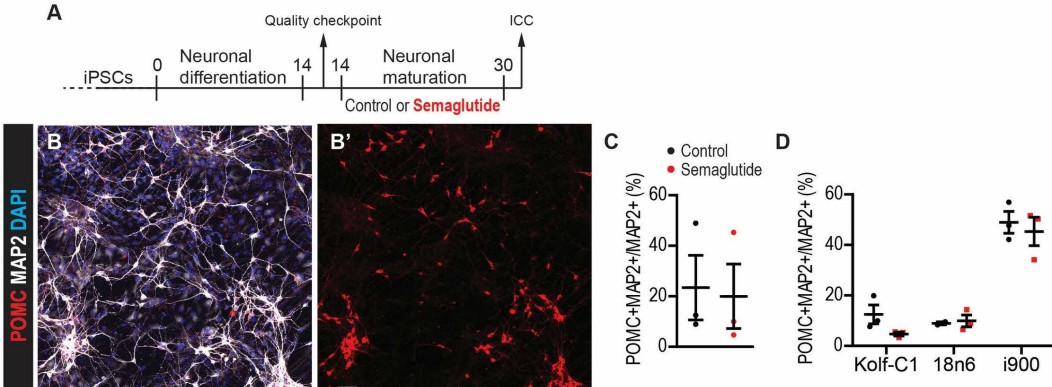

**Fig 3. Semaglutide does not change the proportion of POMC+ hiPSC-derived neurons.** (A) A representative confocal image hiPSC-derived neurons stained as indicated. (B) Quantification of the number of primary neurites per cell. (C) The ratio of the mean pixel density for POMC and MAP2 in hiPSC-derived neurons. (D) The ratio of the mean pixel density for POMC and DAPI in hiPSC-derived neurons. (E) The mean pixel density of POMC in hiPSC-derived neurons. S.b. = 50 µm. n = 3 hiPSC lines. Two-tailed T-Test. Data are presented as mean ± SEM.

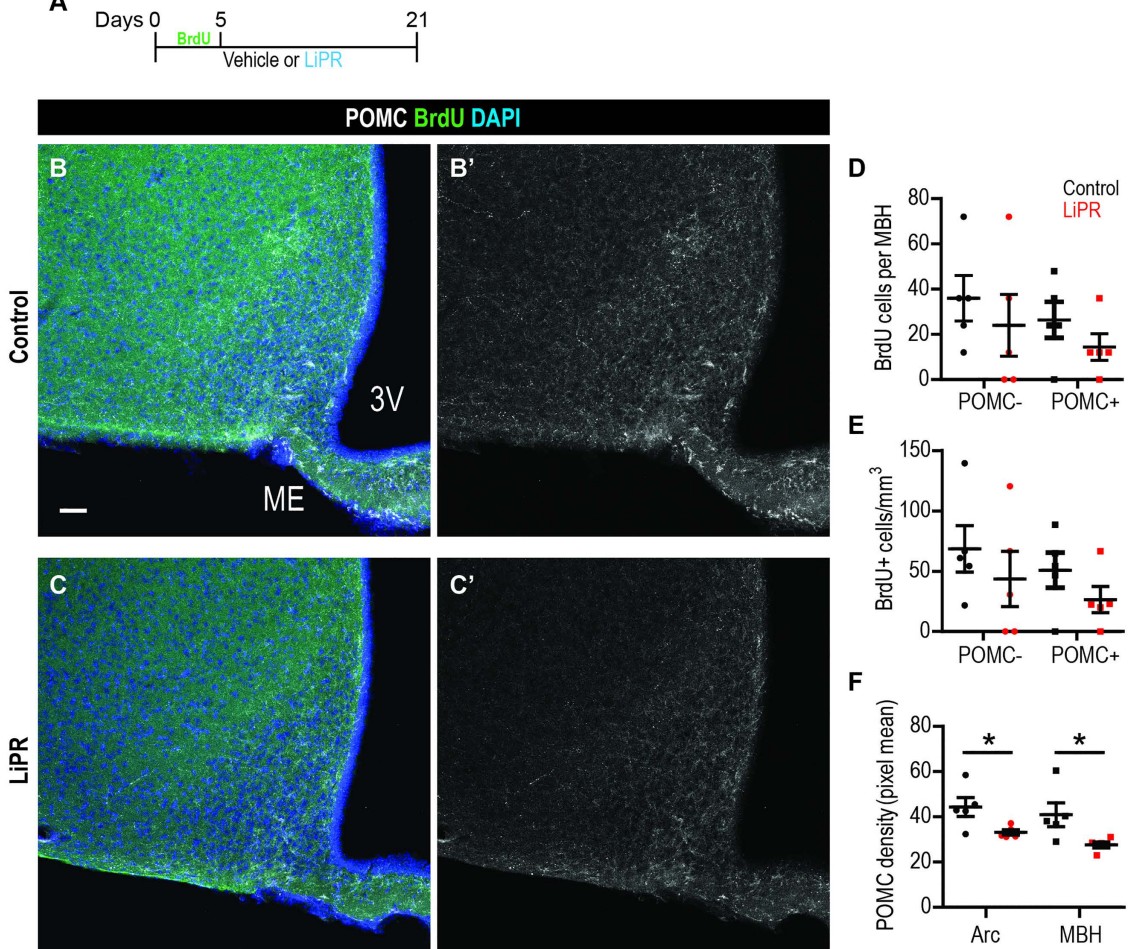

**Fig 4. LiPR does not alter the number or density of new POMC+ neurons in the mouse MBH.** (A) A schematic of the treatment protocol. (B-C) Representative confocal images of the MBH stained as indicated from Control (B) and LiPR treated (C) mice. (D-E) Quantification of the number (D) and density (E) of BrdU+ cells positive or negative for POMC in the MBH. (F) Mean pixel density of POMC in the MBH and Arc. S.b. = 50 μm. n = 5 mice per group. Two-tailed T-Test: *p < 0.05. Data are presented as mean ± SEM.

LiPR did not change the number or proportion of POMC+ neurons in mouse MBH. These findings suggest that a short-term treatment with AOMs in the context of normal diet does not alter the proportion of anorexigenic neurons in the adult hypothalamus.

While we did not observe any change in the proportion or number of POMC+ neurons *in vitro* or *in vivo*, we recognize the limitations of our methodology and approaches. We quantified POMC+ neurons but did not fully address their large phenotypic variability. For example, in adult mice, 27% of POMC+ neurons also express AgRP and NPY [34], suggesting a mixed anorexigenic and orexigenic phenotype. Indeed, specific mouse hypothalamic neurons with negligible levels of POMC (so-called "ghost" neurons) are still anorexigenic [38] and if they are present in our hiPSC-derived cultures, we might not be able to detect them. Similarly, POMC+ neurons vary in their levels of *Cartpt* or *Mc3r* expression [33,34], which can be addressed not by population but by single-cell approaches. And while 18% of mouse neurons highly expressing *Pomc* also express *Prlhr* [33], staining for GPR10 in our hiPSC-derived hypothalamic neurons was not specific and therefore unreliable to assess what proportion of POMC+ neurons also express GPR10.

In our experiments, we exposed hiPSC-derived neurons to AOMs only during their final maturation. While we cannot rule out that exposing cells to AOMs during earlier neuronal differentiation would not alter the proportion of POMC+ neurons, the process of neuronal subtype specification is associated with neuronal maturation as well as fate-specification during early differentiation [39,40]. Importantly, we exposed mouse adult neuronal progenitors to LiPR during both their differentiation and maturation and yet it did not alter the proportion of new POMC+ neurons in the MBH. This may suggest that neuronal phenotypic maturation in the adult hypothalamus may be resistant to environmental interventions. Indeed, exposure to a High-Fat Diet influences hypothalamic adult neurogenesis but not the proportion of new anorexigenic or orexigenic neurons [41,42]. It is also possible that we did not allow enough time for the AOMs to exert their effects during neuronal maturation. The hiPSC protocol we used originally allowed for 22 days of maturation instead of 16 days in our experiments [28,29]. However, even this shorter maturation period generated a higher proportion of POMC+ neurons than originally reported [28], which contests the arguments that it was too short for detecting the effects of AOMs on POMC+ neurons.

Besides the length of the maturation period, there are other technical limitations of our experimental approach. First, we exposed the cells or animals only to a single concentration of PrRP, LiPR or Semaglutide. It is possible that higher concentrations may alter the neuronal subtype specification, however, the concentration of 1 µM used in our cell cultures is two or three orders of magnitude higher than the reported $EC_{50}$ values for all three compounds: 1 nM for PrRP, 0.9 nM for LiPR [14], and 11.2 nM for Semaglutide [43]. Also, 1 µM PrRP was able to stimulate hiPSC-derived hypothalamic neuron activity [26]. Based on this, we argue that the concentration used likely exceeds the $EC_{100}$ for all three compounds and increasing it would likely cause non-specific binding effects. Similarly, the dose of LiPR used in mice (5 mg/kg) has been shown to reduce the body weight, food intake and promote neurogenesis [14,26,44] and is likely sufficient for any neural effects.

Second, we modified the original hiPSC protocol by using one different small molecule. Kirwan et al. used XAV939 (a tankyrase inhibitor) to repress Wnt/β-catenin signaling from day 0 to day 9 [29]. We replaced it with a different tankyrase inhibitor, IWR-1, that is routinely used in our lab to potently repress WNT/β-catenin signaling. Although $IC_{50}$ for tankyrases 1 and 2 differ between XAV939 (11 nM and 4 nM, respectively) and IWR-1 (131 nM and 56 nM, respectively) [45], the range of concentrations we used (0.25–1 µM of IWR-1) is well above the $IC_{50}$ for both tankyrases and match the inhibitory potency of concentrations used by Kirwan et al. (0.5–2 µM of XAV939) [29]. Nevertheless, we recognize that these two inhibitors bind to a different binding subsite on their substrates and that XAV939 has two orders of magnitude higher $IC_{50}$ for PARP1/2 tankyrases than IWR-1 [45].

Third, hiPSC-derived neurons generated in our experiments show expression of both NKX2.1 and FOXG1, which are transcription factors considered mutually exclusive in their expression patterns to determine the development of the diencephalon and telencephalon, respectively [46,47]. Indeed, FoxG1 is essential for telencephalic development, delineates the ventral telencephalic boundary, and is not widely expressed in the diencephalon [46]. However, more recent results show overlapping expression of Nkx2.1 and FoxG1 in two regions of embryonic hypothalamus in mice, in the preoptic area, which form the rostral hypothalamus, and in the boundary of terminal hypothalamus, which develops into the tuberal hypothalamus or MBH [48]. This suggests that both transcription factors are functional in some developing hypothalamic neurons and their co-expression in hiPSC-derived hypothalamic neurons is not aberrant. Indeed, hiPSC-derived hypothalamic neurons in our experiments express POMC, which requires functional NKX2.1 [49]. On the other hand, we recognize that the ratio of NKX2.1 and FOXG1 expression in neurons from our experiments is lower than reported previously despite utilizing the same hiPSC protocol [28]. This difference can be probably attributed in part to technical variability or intrinsic tendencies of different hiPSC lineages towards specific developmental trajectories. However, it is also possible that the lower expression of NKX2.1 in our experiments is a consequence of the concentration of IWR-1 used. Rajamani et al. observed that there are concentration-dependent effects on the proportion of NKX2.1-expressing hiPSC-derived neurons, with 1 µM IWR-1, which we used, generating lower proportions than 10 µM IWR-1 [50].

Despite these technical limitations, our results suggest that a short exposure to AOMs in the context of normal diet does not alter development of POMC-expressing cells. The future work should address whether AOMs can affect the neuronal specification and phenotypic maturation in the context of a High Fat Diet or obesity, during longer exposure, and in hiPSC lines derived from obese or young people.

## Supporting information

**S1 Fig. Proportion of POMC+DAPI+ cells and expression of *NKX2.1* and *FOXG1*.** (A) Quantification of the proportion of POMC+DAPI+/DAPI+ cells of Control and PrRP-treated hiPSC-derived neurons at day 30. (B) Proportion of POMC+DAPI+ cells per hiPSC line (individual data points represent technical replicates). (C) RT-qPCR expression of NKX2.1 and FOXG1 at day 14 and 30 from Control and PrRP-treated hiPSC-derived neurons. n = 3 hiPSC lines. Two-tailed T-Test: **$p < 0.01$, ***$p < 0.001$. Data are presented as mean ± SEM.
(TIF)

## Acknowledgments

The authors thank Prof. Anne White, University of Manchester, for providing the A1H5 antibody, and Prof. Meng Li, Cardiff University, for sharing an aliquot of the i900 hiPSCs. The authors also thank Heather Titterton and Lwin Myint Wai for their technical assistance.

## Author contributions

**Conceptualization:** David Petrik.

**Investigation:** Sara K.M. Jörgensen, May Surridge-Smith, David Petrik.

**Resources:** Lenka Maletínská, Nicholas D. Allen, David Petrik.

**Supervision:** Nicholas D. Allen, David Petrik.

**Validation:** Kimberley Jones.

**Writing – original draft:** David Petrik.

**Writing – review & editing:** Nicholas D. Allen, David Petrik.

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
