## [Decision Letter · Decision Letter 0]

19 May 2025

Dear Dr. Petrik,

Thank you for submitting your manuscript to PLOS ONE. After careful consideration, we feel that it has merit but does not fully meet PLOS ONE’s publication criteria as it currently stands. Therefore, we invite you to submit a revised version of the manuscript that addresses the points raised during the review process.

We look forward to receiving your revised manuscript.

Kind regards,

Alexandra Kavushansky, PhD

Academic Editor

PLOS ONE

Journal Requirements:

2. To comply with PLOS ONE submissions requirements, in your Methods section, please provide additional information regarding the experiments involving animals and ensure you have included details on (1) methods of sacrifice, and (2) efforts to alleviate suffering.

Reviewers' comments:

Reviewer's Responses to Questions

**Comments to the Author**

1. Is the manuscript technically sound, and do the data support the conclusions?

Reviewer #1: Yes

2. Has the statistical analysis been performed appropriately and rigorously?

Reviewer #1: Yes

3. Have the authors made all data underlying the findings in their manuscript fully available?

Reviewer #1: Yes

4. Is the manuscript presented in an intelligible fashion and written in standard English?

Reviewer #1: Yes

Reviewer #1: This study investigated the effects of PrRP, LiPR, and semaglutide on the POMC neuron maturation, but did not find any significant effects of these drugs. The authors need to revise the title because LiPR appears only in Figure 4. They may consider deleting specific drug names or include PrRP as well.

**Do you want your identity to be public for this peer review?** For information about this choice, including consent withdrawal, please see our Privacy Policy

Reviewer #1: No

---

## [Author Response · Author response to Decision Letter 1]

11 Jun 2025

Response to Reviewers

PONE-D-24-58169

Title: “Anti-obesity compounds, Semaglutide and LiPR, do not change the proportion of human and mouse POMC+ neurons”

Journal Requirements:

Answer: We revised the manuscript and adjusted the subheadings, figure references, and text to style requirements and file naming of PLOS ONE.

2. To comply with PLOS ONE submissions requirements, in your Methods section, please provide additional information regarding the experiments involving animals and ensure you have included details on (1) methods of sacrifice, and (2) efforts to alleviate suffering.

Answer: In Methods section, we included details on (1) methods or sacrifice, and (2) efforts to alleviate suffering. The new details are included on lines 261-264 (new text highlighted in red font): “After 21 days of treatment, animals were sacrificed by pentobarbital overdose followed by a complete exsanguination, transcardially perfused with 4% PFA and brain isolated as described before (26). The total anaesthesia was used to nullify suffering.”

Answer: We provided a complete Data Availability Statement in the submission form. We deposited all raw data related to all figures in our manuscript. The data were deposited to a public data depository, OSF, approved by PLOS ONE. The data has following DOI: https://doi.org/10.17605/OSF.IO/84ZXK This DOI is now linked to the manuscript.

4. We note that the grant information you provided in the ‘Funding Information’ and ‘Financial Disclosure’ sections do not match. When you resubmit, please ensure that you provide the correct grant numbers for the awards you received for your study in the ‘Funding Information’ section.

Answer: We provided correct grant numbers to match information provided in the ‘funding information’ and ‘financial disclosure’.

Answer: The data related to the phrase “data not shown” are not a core part of the research being presented and the phrase was removed.

6. Please review your reference list to ensure that it is complete and correct.

Answer: We reviewed our reference list to ensure that it is complete and correct.

Review Comments to the Author:

Reviewer #1: This study investigated the effects of PrRP, LiPR, and semaglutide on the POMC neuron maturation, but did not find any significant effects of these drugs. The authors need to revise the title because LiPR appears only in Figure 4. They may consider deleting specific drug names or include PrRP as well.

Answer: We included PrRP in the manuscript title as suggested by the reviewer. The new title is “Anti-obesity compounds, Semaglutide and LiPR, and PrRP do not change the proportion of human and mouse POMC+ neurons”.

---

## [Decision Letter · Decision Letter 1]

15 Jul 2025

Anti-obesity compounds, Semaglutide and LiPR, and PrRP do not change the proportion of human and mouse POMC+ neurons

PONE-D-24-58169R1

Dear Dr. Petrik,

We’re pleased to inform you that your manuscript has been judged scientifically suitable for publication and will be formally accepted for publication once it meets all outstanding technical requirements.

Kind regards,

Alexandra Kavushansky, PhD

Academic Editor

PLOS ONE
---

## [Editor Report · Acceptance letter]

PONE-D-24-58169R1

PLOS ONE

Dear Dr. Petrik,

I'm pleased to inform you that your manuscript has been deemed suitable for publication in PLOS ONE. Congratulations! Your manuscript is now being handed over to our production team.

Kind regards,

on behalf of

Dr. Alexandra Kavushansky

Academic Editor

PLOS ONE